# Biomarkers for Predicting Response to Personalized Immunotherapy in Gastric Cancer

**DOI:** 10.3390/diagnostics13172782

**Published:** 2023-08-28

**Authors:** Moonsik Kim, Ji Yun Jeong, An Na Seo

**Affiliations:** 1Department of Pathology, School of Medicine, Kyungpook National University, 136-gil 90, Chilgokjungang-daero, Buk-gu, Daegu 41405, Republic of Korea; teiroa83@knuh.kr (M.K.); jjiyun@gmail.com (J.Y.J.); 2Department of Pathology, Kyungpook National University Chilgok Hospital, 807 Hogukno, Buk-gu, Daegu 41404, Republic of Korea

**Keywords:** gastric cancer, immunotherapy, molecular pathology, biomarker, programmed cell death-ligand 1

## Abstract

Despite advances in diagnostic imaging, surgical techniques, and systemic therapy, gastric cancer (GC) is the third leading cause of cancer-related death worldwide. Unfortunately, molecular heterogeneity and, consequently, acquired resistance in GC are the major causes of failure in the development of biomarker-guided targeted therapies. However, by showing promising survival benefits in some studies, the recent emergence of immunotherapy in GC has had a significant impact on treatment-selectable procedures. Immune checkpoint inhibitors (ICIs), widely indicated in the treatment of several malignancies, target inhibitory receptors on T lymphocytes, including the programmed cell death protein (PD-1)/programmed death-ligand 1 (PD-L1) axis and cytotoxic T-lymphocyte-associated protein 4 (CTLA4), and release effector T-cells from negative feedback signals. In this article, we review currently available predictive biomarkers (including PD-L1, microsatellite instability, Epstein–Barr virus, and tumor mutational burden) that affect the ICI treatment response, focusing on PD-L1 expression. We further briefly describe other potential biomarkers or mechanisms for predicting the response to ICIs in GC. This review may facilitate the expansion of the understanding of biomarkers for predicting the response to ICIs and help select the appropriate therapeutic approaches for patients with GC.

## 1. Introduction

Despite advances in diagnostic imaging, surgical techniques, and systemic therapy, gastric cancer (GC) is one of the most common cancers, with a high mortality rate from cancer around the world [1,2]. Unfortunately, morphological and molecular heterogeneity and, consequently, acquired resistance in GC are the major causes of failure in the development of biomarker-guided targeted therapies [2,3]. However, by showing promising survival benefits in some studies, the recent emergence of immunotherapy in GC has had a significant impact on treatment-selectable procedures (Figure 1) [4,5,6,7,8]. To protect the host from external antigens and autoimmune reactions, the immune system is regulated via a number of receptor–ligand interactions [2,9,10]. Immune checkpoint inhibitors (ICIs), widely indicated in the treatment of several malignancies, target inhibitory receptors on T lymphocytes, including the programmed cell death protein (PD-1)/programmed death-ligand 1 (PD-L1) axis and cytotoxic T-lymphocyte-associated protein 4 (CTLA4), and release effector T-cells from negative feedback signals [2,10,11]. Currently, with limited samples, it is important to identify target biomarkers for predicting response to ICIs [12,13]. PD-L1 immunohistochemistry (IHC), microsatellite instability (MSI)/mismatch repair (MMR), Epstein–Barr virus (EBV), and tumor mutational burden (TMB) have been proposed as predictive biomarkers to predict response to ICIs in patients with GC [12,14]. In this article, we review currently available approved predictive biomarkers (excluding EBV) that affect the ICI treatment response, particularly focusing on PD-L1 expression. Unfortunately, the analysis of PD-L1 expression by IHC has several crucial challenges, including inter-observer variation in scoring and the use of different antibodies and staining platforms [13]. We also briefly outline these significant challenges. We further briefly describe alternate promoter utilization as a potential mechanism of resistance to ICIs in GC. This review may facilitate the expansion of the understanding of predictive biomarkers for ICIs and help select the appropriate therapeutic approaches for patients with GC.

## 2. PD-L1 Expression as a Biomarker in GC

### 2.1. Rationale and Performance

Honjo’s group first discovered PD-1 in 1992 [11,15]. PD-1 is mainly expressed in activated cytotoxic T-cells and other immune cells [5,11,16]. It is a cell surface (or transmembrane) protein encoded by the CD274 gene [2,17]. The interaction of PD-1, expressed on cytotoxic T lymphocytes, with PD-L1 on antigen-presenting cells is one of the main mechanisms of immune modulation, thereby allowing T-cell inactivation and tumor immune evasion [2,10,16,18]. Tumor cells with mutable neoantigens are recognized as ‘non-self’ by the immune system and are selectively targeted and removed by inducing an immune response [10]. To avoid removal, tumor cells can upregulate PD-L1 and PD-L2 expression following exposure to interferon-γ and other signaling and cytokines [10,11,16,17,19]. Furthermore, PD-L1 expression is increased in some immune cells within the tumor microenvironment, including dendritic cells, macrophages, antigen-presenting cells, and T-cells [17,20]. By this principle, the blockade of PD-1/PD-L1 interaction by monoclonal antibodies against either PD-1 (pembrolizumab and nivolumab) or PD-L1 (durvalumab, atezolizumab, and avelumab) appears to be a logical therapeutic strategy, particularly for a highly antigenic solid tumor, including GC [10,21]. 

The phase 2 KEYNOTE-059 trial (NCT#02335411), which enrolled 259 patients with locally advanced or metastatic gastric or gastroesophageal junction (G/GEJ) adenocarcinoma suggested the safety and effectiveness of pembrolizumab as a third-line treatment [4,11]. In this single-arm, multicohort trial, the objective response rate (ORR) was 11.6% (30 of 259 patients), and complete responses (CRs) were noted in 2.3% of patients (6 of 259); the median (range) response duration was 8.4 (1.6+–17.3+) months [4,11]. Among 259 patients, 148 (57.1%) with PD-L1-positive tumors (combined positive score [CPS] ≥1, evaluated using the PD-L1 22C3 pharmDx Kit) had ORRs and CRs of 15.5% (23 of 148) and 2.0% (3 of 148), respectively. The median response duration was 16.3 (1.6+–17.3+) months. Based on these results, pembrolizumab, in 2017, was granted accelerated Food and Drug Administration (FDA)-approval as a third-line therapy for patients with locally advanced or metastatic G/GEJ adenocarcinoma whose tumors express a CPS of ≥1 using the PD-L1 immunohistochemistry (IHC) 22C3 pharmDx assay [2,18]. Notably, pembrolizumab was approved, along with a companion diagnostic test, the PD-L1 IHC 22C3 pharmDx assay, for use on the Dako Autostainer Link 48 platform [4,18]. However, after accelerating FDA approval, pembrolizumab failed to demonstrate significant survival improvements in the following phase 3 trials (KEYNOTE-061 [NCT#02370498; efficacy as a second-line therapy] and KEYNOTE-062 [NCT#02494583; efficacy as a first-line therapy]) in patients with advanced G/GEJ adenocarcinoma showing a PD-L1 CPS of ≥1 [5,6,11,18]. 

Another phase 3 CheckMate-649 trial (NCT#02872116), which enrolled 1581 patients with untreated, unresectable, Human Epidermal Growth Factor Receptor 2 (HER2)-negative G/GEJ, or esophageal adenocarcinoma, demonstrated the acceptable safety profile and efficacy of nivolumab as a first-line treatment [7,18]. The CheckMate-649 trial demonstrated significant improvements in the overall survival (OS) and progression-free survival (PFS) for patients with a CPS of ≥5 (955 of 1581; 60.4%) determined using the PD-L1 28-8 pharmDx kit on the Autostainer Link 48 platform [7]. Specifically, the median OS was 14.4 months (95% confidence interval [CI]: 13.1–16.2) in the nivolumab plus chemotherapy group (n = 789) versus 11.1 months (95% CI: 10.0–12.1) in the chemotherapy-alone group (n = 792) (*p* < 0.0001) [7]. Moreover, the median PFS was 7.7 months (95% CI: 7.0–9.2) in the nivolumab plus chemotherapy arm versus 6.0 months (95% CI: 5.6–6.9) in the chemotherapy alone arm (*p* < 0.0001) [7]. Based on these results, in 2021, the FDA-approved, nivolumab plus fluoropyrimidine- and platinum-based chemotherapy as a first-line therapy for HER2-negative advanced or metastatic G/GEJ and esophageal adenocarcinoma [2]. Nivolumab was approved along with a companion diagnostic test, the PD-L1 IHC 28-8 pharmDx assay, for use on the Dako Autostainer Link 48 platform [2,18]. However, the phase 3 ATTRACTION-4 trial demonstrated that first-line nivolumab plus oxaliplatin-based chemotherapy resulted in a significant improvement in PFS, but not OS, in Asian patients with untreated, HER2-negative, unresectable advanced or recurrent G/GEJ adenocarcinoma with a tumor proportion score (TPS) of ≥1 (114 of 724; 15.7%) [22]. In this trial, PD-L1 expression was evaluated using the PD-L1 IHC 28-8 pharmDx assay and was scored using TPS, not CPS [22]. 

Recently, the interim results of the phase 3 KEYNOTE-811 trial (NCT#03615326) in enrolled patients with unresectable or metastatic, HER2-positive G/GEJ adenocarcinoma showed that first-line pembrolizumab plus trastuzumab and chemotherapy significantly reduced tumor size, induced CRs in some participants, and significantly improved ORR [8]. In this trial, PD-L1 expression was evaluated using the PD-L1 IHC 22C3 pharmDx assay and was scored using CPS (patients with a CPS of ≥1: 582 of 692, 84.1%) [8]. Although KEYNOTE-811 trial has proceeded, pembrolizumab plus trastuzumab and chemotherapy could potentially be a new first-line treatment option for patients with unresectable advanced or metastatic, HER2-positive G/GEJ adenocarcinoma. Table 1 summarizes the clinical trials mentioned so far. 

### 2.2. Interpretation of PD-L1 IHC Assays in GC

To date, it has been typical to carry out IHC for evaluating PD-L1 expression [17]. When assessing PD-L1 expression, pathologists must pay attention to reproducibility and accuracy [17]. Criteria vary depending on the type of tumor. However, both the PD-L1 IHC 22C3 pharmDx and PD-L1 IHC 28-8 pharmDx assays share the CPS scoring system for assessing PD-L1 expression [2,18]. CPS is quantified as the number of PD-L1-stained cells (tumor cells, lymphocytes, and macrophages) and dividing the result by the total number of viable tumor cell subsequently multiplied by 100 [2,11,18]. For proper evaluation, at least 100 viable tumor cells must be present in the PD-L1-stained slide, and the maximum score is defined as 100 if the calculated results exceed 100 [2,11,18]. In CPS scoring system, partial linear or complete circumferential membrane staining (at any intensity) of invasive viable tumor cells is assessed, but not dysplasia or in situ carcinoma [2,11,18]. Conversely, membrane and/or cytoplasmic staining (at any intensity) of mononuclear inflammatory cells (including lymphocytes and macrophages) within tumor nests and adjacent supporting stroma is evaluated, but not eosinophils, neutrophils, and plasma cells. Additionally, normal cells, stromal cells (including fibroblasts), and cellular debris and/or necrotic cells are excluded from the numerator [18]. If the PD-L1 staining pattern shows heterogeneity, the final CPS should be determined by assessing the CPS results for each area within the entire tumor [18]. As previously mentioned, the interpretation of PD-L1 positivity should be according to the CPS cutoff point suitable for the assays used in the evaluation since two different PD-L1 assays have been approved as companion diagnostic assays on the basis of different CPS cutoff points [18]. CPS positivity is CPS ≥ 1 and CPS ≥ 5 in the PD-L1 IHC 22C3 pharmDx and PD-L1 and IHC 28-8 pharmDx assays, respectively [2,18]. In HER2-negative patients, the National Comprehensive Cancer Network Guidelines (NCCN) guidelines for GC version 2.2022 recommend fluoropyrimidine (fluorouracil or capecitabine), oxaliplatin, and nivolumab (CPS ≥ 5 when using the PD-L1 IHC 28-8 pharmDx assay) as preferred regimens of first-line therapy [14]. 

Another scoring method for PD-L1 IHC expression in other solid tumors, including metastatic non-small-cell lung cancer and melanoma, uses TPS for calculating the percentage of stained tumor cells out of the total tumor cells [2]. However, as TPS does not contain tumor-infiltrating immune cells when calculating the score, it may be inefficient in identifying responders [2,23]. 

Interestingly, a new tumor area positivity (TAP) score has recently been proposed for the evaluation of the VENTANA PD-L1 SP263 assay in G/GEJ adenocarcinoma and esophageal squamous cell carcinoma [24]. The TAP scoring system is measured as the percentages of the PD-L1-positive tumor cells plus immune cells are divided by the tumor area, which is occupied by all viable tumor cells and the tumor-associated stroma containing tumor-associated immune cells [24]. Partial linear or circumferential membrane staining (at any intensity) of tumor cells is evaluated, and cytoplasmic, membranous, and punctate staining of tumor-associated immune cells (at any intensity) is considered PD-L1-positive staining [24]. Liu et al. reported that the degree of agreement between the TAP (cutoff of 1%) and CPS (cutoff of 1%) was 100% positive percent agreement, 84.6% negative percent agreement, and 96.2% overall percent agreement [24]. Furthermore, they also suggested that the TAP scoring system is a simple, visual-based method as the average time spent on scoring is 5 min, and it can address the limitations of a cell-counting approach [24]. However, accumulating evidence based on clinical trials is required for the standardization of this scoring system.

### 2.3. Interchangeability of PD-L1 Assays in GC

The interchangeability between the PD-L1 IHC 22C3 pharmDx and the PD-L1 IHC 28-8 pharmDx assays is very high [25,26]. Ahn et al. compared two PD-L1 assays in the same formalin-fixed, paraffin-embedded (FFPE) tissue blocks from 55 cases with GC and demonstrated that all PD-L1-positive cases using the PD-L1 IHC 22C3 pharmDx assay were also positive using the PD-L1 IHC 28-8 pharmDx assay, regardless of which CPS cutoff was used (≥1, ≥10, and ≥50) [25]. Moreover, they provided a high correlation between them in a comparison of the quantitative CPS of the two assays (Spearman correlation value = 0.978, *p* < 0.001) [25]. However, nonspecific background staining can be observed in the PD-L1 IHC 28-8 pharmDx assay. When the PD-L1 IHC 28-8 pharmDx assay was used in comparison with the PD-L1 IHC 22C3 pharmDx assay, nonspecific cytoplasmic staining, but not membranous staining, was observed in the muscular tissues and tumor cells [25]. Moreover, Narita et al. evaluated a pairwise comparison of two assays in the same FFPE tissue microarray (TMA) from 226 cases with GC. They reported that 87% of the pairs were concordant, and 11% had a higher expression using the PD-L1 IHC 22C3 pharmDx assay [26]. With a CPS of ≥ 5, the concordance between them was strong (kappa score = 0.881), specifically, 25 and 22 cases were positive using the PD-L1 IHC 22C3 pharmDx and PD-L1 IHC 28-8 pharmDx assays, respectively [26]. Conversely, Yeong et al. suggested that CPSs using the PD-L1 IHC 28-8 pharmDx assay were consistently observed at higher rates than that using the PD-L1 IHC 22C3 pharmDx assay in 362 cases with GC (344 cores of TMA, 18 whole slides) (70.3% versus 49.4%, *p* < 0.001 at CPS ≥ 1; 29.1% versus 13.4%, *p* < 0.001 at CPS ≥ 5; 13.7% versus 7.0%, *p*  =  0.004 at CPS ≥ 10) [27]. However, in their study, PD-L1 expression was determined by multiplex IHC/immunofluorescence using the Opal Multiplex fIHC kit (Akoya Biosciences, CA, USA) [27]. Moreover, Kim et al. demonstrated that the concordance for the different IHC assays (VENTANA PD-L1 SP263, PD-L1 IHC 22C3 pharmDx on the Dako automated staining platform, and PD-L1 IHC 22C3 pharmDx on the Ventana platform) was very low across all cutoffs in the biopsy or resection specimens (biopsy, kappa coefficient [κ] = 0.17–0.453; resection, κ = 0.02–0.311) [28]. Therefore, based on the results of several studies, even when using the same sample, PD-L1 expression can be observed differently depending on the antibody, staining method, and platform/machine [25,26,27,28,29]. 

### 2.4. Discordance between Biopsy and Resection Specimens and Inter-Observer Variation

As previously mentioned, as PD-L1 staining patterns may exhibit intratumoral heterogeneity, there may be discrepancies in PD-L1 CPS results between paired biopsy and resection specimens [28,29]. Kim et al. reported that the overall positive agreement for PD-L1 results, when the CPS cutoff was 1, in paired biopsy and resection samples from 99 cases with GC was 100% (VENTANA PD-L1 SP263; κ = 1.000), 86% (PD-L1 IHC 22C3 pharmDx on the Dako automated staining platform; κ = 0.693) and 93% (PD-L1 IHC 22C3 pharmDx on the Ventana platform; κ = 0.820), respectively [28]. Additionally, Heo et al. presented that cases with PD-L1 CPS ≥1 were observed in 32.1% (36 of 112) paired biopsy and 47.3% (53 of 112) resection samples as measured by digital image analysis using the PD-L1 IHC 22C3 pharmDx assay [29]. Interestingly, in their study, 41 (36.6%) discrepant cases between biopsy and resection were determined by digital image analyses (κ = 0.254, *p* = 0.0048), while 31 (27.7%) discrepant cases were determined by pathologists (κ = 0.432, *p* < 0.0001) [29]. Furthermore, Yamashita et al. showed that cases with PD-L1 CPS ≥ 1 observed in 46.6% (89 of 191) paired biopsy and 70.1% (135 of 191) resection samples using the E1L3N antibody for PD-L1 [30]. In their study, the accordance of cases with only a single biopsy tissue was significantly lower (48.8%) than that of cases with multiple biopsied tissues (68.9%) (*p* < 0.05) [30]. Therefore, owing to the high levels of intraumoral and intertumoral heterogeneity in GC, PD-L1 expression in paired biopsy and resection specimens of GC may show relatively low concordance. Considering the difference in interpretation between digital image analyses and pathologists, inter-observer variation may also exist. Multiple biopsies can be more helpful than a single biopsy for reducing discrepancies in PD-1 CPS results between biopsy and resection [30]. Notably, Schoemig-Markiefka et al. suggested that sampling and analysis of four or more biopsies with a total area of approximately 4.5 mm^2^ may obtain similar results to resection specimens [31]. To evaluate inter-observer variation, Park et al. compared the CPS results of 55 cores of TMA obtained from five pathologists [32]. The PD-L1 IHC 22C3 pahrmDx assay had a higher concordance among pathologists than the VENTANA PD-L1 SP263 assay (CPS ≥ 1, κ = 0.389 versus κ = 0.224; CPS ≥ 10, κ = 0.256 versus κ = 0.140) [32]. They suggested that continuous training for PD-L1 interpretation is significant as trained pathologists showed higher agreement between the two assays than untrained pathologists [32].

## 3. Other Biomarkers Associated with the Immune Microenvironment (IME) and Immunotherapy in GC

### 3.1. MSI/MMR-Deficiency (MMR-D)

MMR genes, including mutL homolog 1 (MLH1), mutS homolog 2 (MSH2), PMS1 homolog 2 (PMS2), and mutS homolog 6 (MSH6), operate in DNA repair pathways in healthy cells. However, the loss of function of these gene products leads to MMR-D [33]. The molecular phenomenon of MMR-D is MSI, with expansion or contraction of the number of short tandem repeats, or microsatellites, throughout the genome [34]. Tumors with microsatellite instability-high (MSI-H) have a higher mutational burden and higher numbers of tumor-infiltrating lymphocytes (TILs) than those with microsatellite stability [35,36,37]. The high mutational burden in tumors with MSI-H results in the production of neopeptides that are presented as antigens in the tumor [38,39], and increased neoantigen production can lead to vigorous immune reaction [38]. In 2017, FDA approved pembrolizumab in patients with MSI-H or MMR-D solid tumors regardless of tumor location (“tumor agnostic”) following failure of prior standard treatment as a second- or subsequent-line treatment [40]. However, recent data show that the rate of response to pembrolizumab in tumors with MSI-H/MMR-D significantly varies depending on the indications [18,41,42]. Notably, post-hoc analysis of phase 2 KEYNOTE-059 and phase 3 KEYNOTE-061 and KEYNOTE-062 trials showed that MSI-H can be a predictive biomarker for pembrolizumab in patients with advanced G/GEJ adenocarcinoma, in spite of the line and treatment received [2,43]. 

GC with MSI-H is one of The Cancer Genome Atlas (TCGA) subtypes and has distinct clinicopathological and molecular characteristics [2,44]. GC with MSI-H is observed in 7–23% of patients with sporadic GC [2,18,44,45,46,47]. The following are the clinical characteristics of GC with MSI-H: (1) gastric antrum, (2) more female, (3) a relatively older age, (4) Lauren intestinal type, (5) early stage, and (6) and relatively favorable prognosis [2,18,44,45]. GC with MSI-H features the gastric CpG island methylator phenotype (CIMP) with MLH1 silencing [2,18,44]. The NCCN guidelines declare that universal testing for MSI using polymerase chain reaction (PCR)/next generation sequencing (NGS) or MMR using IHC should be carried out for all patients with newly diagnosed GC in accordance with the College of American Pathologists DNA Mismatch Repair Biomarker Reporting Guidelines [2,14,18]. To measure the gene expression levels of microsatellite markers, the MSI status is evaluated using PCR (Bethesda/National Cancer Institute panel: D2S123, D17S250, D5S346, BAT-25, and BAT-26; or NR-21, NR-24, NR-27 [or Mono-27], BAT-25, and BAT-26) [2,14,18]. MMR-D is assessed using IHC to directly evaluate the nuclear expression of proteins (MLH1, MSH2, PMS2, and MSH6) [2,14,18]. The NCCN guidelines are recommended to be performed in a Clinical Laboratory Improvement Amendment-approved laboratory when the tissues available for testing are limited or when a patient cannot undergo a traditional biopsy. This is because the use of sequential testing of single biomarkers or limited molecular diagnostic panels can quickly deplete the samples [14]. 

Interestingly, in more than 90% of GC with MSI-H, MMR-D shows MLH1 and/or PMS2 losses owing to the hypermethylation of the MLH1 gene [2,14,18]. Although MSI using PCR/NGS and MMR-D using IHC show similar performance characteristics and high concordance rate (>90%) [18], they provide fundamentally different information. Therefore, their co-testing can increase the overall number of correctly characterized tumors by more than 99% [2]. However, a recent clinical trial (NCT#02589496) demonstrated that patients with metastatic GC with MSI-H had an overall response rate of 85.7% to pembrolizumab. However, in one unresponsive patient, intratumoral heterogeneity of the MSI status was observed [48]. 

### 3.2. EBV 

EBV is a gamma herpes virus, and it is now widely known to cause neoplasms of various cell origins, including nasal NK/T-cell lymphoma, classic Hodgkin lymphoma, Burkitt lymphoma, nasopharyngeal carcinoma, GC, and leiomyosarcoma [2,49]. EBV-associated gastric cancer (EBVaGC) comprises approximately 10% of global GCs, with variable morbidity between ethnicity and geographic regions [2,18]. As one of the TCGA subtypes, EBVaGC has distinct clinicopathological, genetic, and IME characteristics [2,18,44]. The following are the clinical characteristics of EBVaGC: (1) higher morbidity in males, (2) occurrence at a relatively younger age, (3) fundus and body (proximal) location of the stomach, (4) prominent TILs, (5) early stage, and (6) and relatively favorable prognosis [2,18]. Gastric carcinoma with lymphoid stroma (GCLS) mentioned in WHO classification significantly correlated with EBV infection, which accounts for 20–90% of GCLSs [2]. However, GCLS can also be observed in GC with MSI, and can be noted in GC with neither EBV nor MSI [2]. EBVaGC exhibits extreme CIMP owing to cyclin-dependent kinase inhibitor 2A promoter hypermethylation and deficiency of MLH1 promoter hypermethylation [2,44,50]. Consequently, EBV and MSI are mutually exclusive in GC [2]. EBVaGC displays frequent mutations in phosphatidylinositol-4,5-bisphosphate 3-kinase, catalytic subunit alpha (PIK3CA) and AT-rich interactive domain-containing protein 1A, and rare mutations in TP53, and the overexpression of PD-L1/2 and interferon-γ [2,18,44,45,46,47]. Interestingly, PIK3CA mutations in EBVaGC exhibited a more distributed pattern, unlike those typically confined to the kinase domain (exon 20) in EBV-negative GC [2,44]. Immune response gene deregulation and PD-L1/2 overexpression provide a rationale for testing ICIs as treatment for patients with EBVaGC [2]. Notably, a recent clinical trial (NCT#02589496) demonstrated that patients with metastatic EBVaGC showed a significant response to pembrolizumab (overall response rate, 100%) [48]. 

Although EBV can be detected using various methods, in situ hybridization (ISH) of EBV-encoded small RNAs (EBERs) is the gold standard for the identification of EBV infection in formalin-fixed, paraffin-embedded tissue blocks [2,18,49]. Intratumoral heterogeneity (juxtaposition of EBER-negative and -positive tumor areas) of EBER-positivity has been reported since the EBER ISH assay can identify EBV-infected cells on histologic sections [2,51,52]. Böger et al. reported that in 484 German patients who underwent surgical resection of GC, 4 (18.2%) of 22 patients with EBVaGC showed heterogenous EBER positivity [51]. Conversely, Kim et al. reported that in 3499 Korean patients who underwent surgical resection of GC, 4 (1.9%) of 214 patients with EBVaGC had EBV heterogeneity [52]. They demonstrated that EBV-positive and-negative regions within tumors exhibited subtly different characteristics, including histological pattern, tumor IME (TIME), and genomic profiles [52].

### 3.3. TMB

TMB is a measure of the total amount of nonsynonymous somatic coding mutations per megabase (Mb) of genome sequenced in a tumor and has been considered a new predictive biomarker for ICI treatment response [2,11,18,38,53]. Specifically, accumulating data have shown that high TMB, or high neoantigen load, is more likely to be associated with a good clinical response to ICIs and improved prognosis [11,17,54]. Whole exome sequencing (WES) is considered optimal for evaluating TMB measurements since the concept of TMB has been initially derived from WES. However, disadvantages, including the high cost, long working time, extensive analysis, and difficult data management, limit the widespread use of WES in daily clinical practice [11,17]. Furthermore, since most available tests using WES require at least 150–200 ng of genomic DNA, limited amounts of DNA in cytological and small biopsy samples can occasionally be problematic [11]. Recently, some commercially available targeted next-generation sequencing (NGS) panels also offer TMB measurements showing clinically high compatibility results with WES [2,38]. Therefore, owing to lower sequencing cost, shorter turn-around time, and lower DNA input amounts in clinical settings, targeted NGS with comprehensive gene panels can be desirable [17,55]. To determinate TMB, it is usually recommended the use of gene panels larger than 1 Mb, or 300 gene, and the standardization of the bioinformatic processes [2,17,53,56,57,58]. In addition, for a reliable TMB assessment, it is important to consider the DNA input amount and/or DNA quality and the percentage of tumor cells (requirement ≥ 20%) in the sample [12,13]. However, there are still several limitations to the adoption of TMB as a predictive biomarker in clinical practice: (1) difficulty of consensus in panel-based TMB quantification, (2) lack of an appropriate way to transform TMB estimates across different panels, and (3) inconsistency of robust cutoff values for TMB [2,18,53]. Notably, the use of formalin-fixed, paraffin-embedded tissues in NGS tests may increase DNA sequence artifacts [17,59]. These sequence artifacts may be caused by formalin fixation as formalin may result in various cross-links, DNA fragmentation, denaturation, and deamination of cytosine bases [17]. Based on the KEYNOTE-158 trial in 2020, FDA approved pembrolizumab in patients with TMB-high (≥10 mutations/Mb) with unresectable or metastatic solid tumors that have progressed following previous treatment and have no satisfactory alternative treatment options [2,14,60]. The prespecified exploratory analysis of the KEYNOTE-062 trial (ClinicalTrials.gov, NCT02494583) provided a relationship between TMB, which assessed by NGS using FoundationOne CDx (cutoff point: ≥10 mut/Mb), and clinical usefulness of first-line pembrolizumab ± chemotherapy in patients with advanced GC. However, they suggested that when MSI-H is excluded, the clinical usefulness of TMB may be undermined [2,18,61]. To date, commercially available panels include the MSK-IMPACT panel (468 genes, 1.22 Mb of the genome), FoundationOne CDx (324 genes, 1.2 Mb; Foundation Medicine, Cambridge, MA, USA), Oncomine Tumor Mutation Load Assay (409 oncogenes, 1.65 Mb; Thermo Fisher Scientific, Waltham, MA, USA), and TruSight Oncology 500 (523 genes for DNA and 55 genes for RNA, 1.94 Mb panel size; Illumina, San Diego, CA, USA) [17].

## 4. Potential Predictive Biomarkers

Recently, somatic recruitment of alternate promoters in GC has been demonstrated as a tumor immune editing mechanism [62,63]. In particular, the reduced production of high-affinity major histocompatibility complex class I binding GC peptides through the loss of immunogenic N-terminal peptides has been suggested as an immune evasion, thereby enabling early tumor formation [62]. However, Sundar et al. demonstrated that metastatic GCs with high alternate promoter utilization exhibited reduced T-cell cytolytic activity markers, lower response rates to anti-PD1 ICIs, and lower PFS [62]. They suggested that a significant proportion of metastatic GCs use alternative promoters as an immune evasion mechanism and that these tumors may be resistant to ICIs [62]. 

## 5. Discussion

We reviewed the prospect and existing limitations of PD-L1 IHC assays and other approved predictive biomarkers (excluding EBV) for ICI response performed using samples from patients with GC (Table 2). The relationship between PD-L1 positivity, MSI-H, EBV positivity, and TMB is depicted in Figure 2 [64]. Unfortunately, it seems clear that no single test can be used as a reproducible proxy for predicting the advantage of immunotherapy [10]. Therefore, developing an integrated predictive model that considers the complex components affecting host–TIME interactions by reflecting the heterogeneity of GC is necessary. However, owing to companion diagnosis of ICIs, pathologists will be regularly requested for PD-L1 IHC assay results by oncologists in daily practice. Therefore, to overcome the inter-observer variation, pathologists should be properly trained using suitable GC samples. Furthermore, automated digital image analysis based on the accumulated technology can help evaluate PD-L1 IHC assays with high accuracy and consistency. Moreover, pathologists and clinicians should remember that MSI and EBV can exhibit intratumoral heterogeneity since this heterogeneity may affect the response to ICIs. Finally, owing to the difficulty of identifying the appropriate cutoff point for high and low TMB, using TMB as a robust predictor of ICI response in daily practice may be challenging.

## 6. Conclusions

Despite various obstacles, advanced technology and genomics have led to the development of personalized and precision medicine. Combinations of approved biomarkers will allow optimal immunotherapy strategies to be selected. Currently, several clinical trials are also underway to develop next-generation ICIs targeting checkpoint regulators beyond PD-1/PD-L1. If these clinical trials are successful, developed drugs will allow patients to select the optimal individual treatment strategy. Therefore, a comprehensive understanding of TIME is becoming more significant. 

## Figures and Tables

**Figure 1 diagnostics-13-02782-f001:**
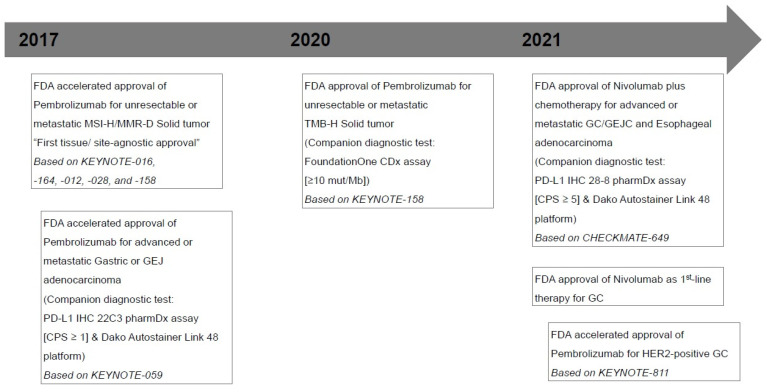
Timeline of major milestones for immunotherapy and U.S. Food and Drug Administration (FDA)-approved Immune checkpoint inhibitors in gastric cancers. MSI-H, microsatellite instability-high; MMR-D, mismatch repair deficiency; GEJ, gastroesophageal junction; PD-L1, programmed death ligand 1; CPS, combined positive score; TMB-H, tumor mutational burden-high; mut/Mb, mutations/megabase; GC, gastric cancer; GEJC, gastroesophageal junction cancer; HER-2, human epidermal growth factor receptor 2.

**Figure 2 diagnostics-13-02782-f002:**
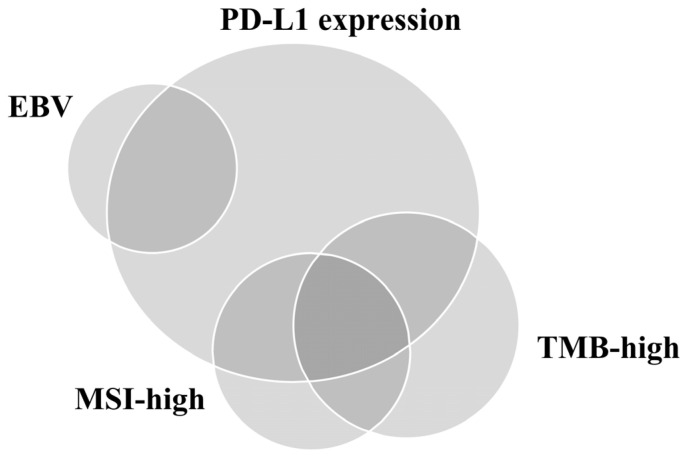
Relationship of programmed death-ligand 1 (PD-L1) expression, Epstein–Barr virus (EBV), microsatellite instability (MSI), and tumor mutational burden (TMB) in gastric cancer.

**Table 1 diagnostics-13-02782-t001:** Summary of clinical trials of immunotherapy in gastric/gastroesophageal junction (G/GEJ) adenocarcinoma.

Trials	Clone	Study Design	Number of Patients	Efficacy Findings
Pembrolizumab				
KEYNOTE-059		phase 2, global, open-label, single-arm, multicohort	259 (148 with CPS ≥ 1)	Objective response rate (ORR) (CR + PR) 11.6%
[4]		Recurrent or metastatic G/GEJ cancer		Median response duration 8.4 (1.6+ to 17.3+)
		pembrolizumab, 200 mg, 3rd-line		Median PFS 2.0 months
				Median OS 5.6 months
KEYNOTE-061		phase 3, global, randomized, open-label, multicohort	592 (395 with CPS ≥ 1)	Pembrolizumab vs. Paclitaxel
[5]	PD-L1 IHC 22C3	Advanced/unresectable or metastatic G/GEJ cancer		Median PFS 1.5 vs. 4.1 months
	pharmDx assay	pembrolizumab, 200 mg, 2nd-line		Median OS 9.1 vs. 8.3 months
KEYNOTE-062	CPS ≥ 1	phase 3, global, randomized, controlled, partially blind	763 with CPS ≥ 1	Pembrolizumab vs. Chemotherapy (CTx)
[6]		Advanced/unresectable or metastatic G/GEJ cancer	(281 with CPS ≥ 10)	Median OS 10.6 vs. 11.1 months
		pembrolizumab, 200 mg, 1st-line		Median PFS 2.0 vs. 6.4 months
				Pembrolizumab + CTx vs. CTx
				Median OS 12.5 vs. 11.1 months
				Median PFS 6.9 vs. 6.4 months
KEYNOTE-811		phase 3, global, randomized, placebo-controlled, double-blind	692 (582 with CPS ≥ 1)	Pembrolizumab vs. Placebo
[8]		HER2-positive unresectable or metastatic G/GEJ cancer		ORR 74% vs. 52% (*p* < 0.0001)
		pembrolizumab, 200 mg, 1st-line		Median duration of response 10.6 vs. 9.5 months
		or placebo (normal saline or dextrose)		
Nivolumab				
CheckMate-649	PD-L1 IHC 28-8	phase 3, randomized, open-label, multicenter	1581 (955 with CPS ≥ 5)	Nivolumab + CTx vs. CTx (patients with CPS ≥ 5)
[7]	pharmDx assay	Untreated, unresectable, HER2-negative		Median OS 14.4 vs. 11.1 months (*p* < 0.0001)
	CPS ≥ 5	G/GEJ or esophageal adenocarcinoma		Nivolumab + CTx vs. CTx (All randomized patients)
		Nivolumab, 360 mg or 240 mg, 1st-line		Median OS 13.8 vs. 11.6 months (*p =* 0.0002)
ATTRACTION-4	PD-L1 IHC 28-8	phase 2-3, randomized, double-blind, placebo-controlled,	724 (114 with TPS ≥ 1)	Nivolumab + CTx vs. CTx
[22]	pharmDx assay	multicenter across Japan, South Korea, and Taiwan		Median PFS 10.45 *versus* 8.34 months (*p* = 0.0007)
	TPS ≥ 1	HER2-negative, unresectable, advanced or recurrent		Median OS 17.45 vs. 17.15 months (*p* = 0.26)
		G/GEJ cancer		
		Nivolumab, 360 mg, 1st-line		

Abbreviation: CPS, combined positive score; CR, complete response; PR, partial response; PFS, progression-free survival; OS, overall survival; TPS, tumor proportion score.

**Table 2 diagnostics-13-02782-t002:** Summary table of representative tests of predictive biomarkers in immunotherapy of gastric cancer.

Predictive Markers	Comments
PD-L1 testing	
Immunohistochemistry (Clone)	Platform: Autostainer Link 48
22C3	Pembrolizumab (Companion diagnostic)
28-8	Nivolumab (Companion diagnostic)
MSI testing	
Immunohistochemistry	Four MMR proteins: MLH1, MSH2, PMS2, and MSH6
	MMR-D is determined in the absence of nuclear expression of at least one MMR protein.
Polymerase chain reaction	Bethesda/National Cancer Institute panel:
	Two mononucleotide (BAT-25 and BAT-26) & Three dinucleotide (D5S346, D2S123, and D17S250)
	Five poly-A mononucleotide panel:
	NR-21, NR-24, NR-27 [or Mono-27], BAT-25, and BAT-26
	The five poly-A panel shows higher sensitivity and specificity.
	MSI-H is determined as instability of two or more of five microsatellite loci.
Next-generation sequencing (NGS)	The major advantage of NGS is that MSI analysis and TMB determination can be performed simultaneously.
EBV testing	
In situ hybridization	It is the most suitable and widely used method to identify EBV in FFPE specimens.
TMB testing	
Whole exome sequencing	Optimal for evaluating TMB measurements
Targeted sequencing	Lower sequencing cost, shorter turn-around time, and lower DNA input amounts

Abbreviation: PD-L1, programmed death ligand 1; MSI, microsatellite instability; MMR, mismatch repair; MMR-D, MMR-deficiency; MSI-H, MSI-high; TMB, tumor mutational burdern; EBV, Epstein–Barr virus; FFPE, formalin-fixed, paraffin-embedded.

## Data Availability

Not applicable.

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
