# Peer review of "Biomarkers for Predicting Response to Personalized Immunotherapy in Gastric Cancer"

_diagnostics, 2023, doi:10.3390/diagnostics13172782_

Round 1

Reviewer 1 Report

This manuscript aims to describe or identify biomarkers for GCs that could be potentially helpful for personalized immunotherapy. However, the main focus of this manuscript was not on the derivation of biomarkers or analysis based on previously established datasets (expression or mutational profiles).

If the authors' main conclusion was to identify novel biomarkers, then they should re-organize the whole manuscript to highlight their analysis. Instead, the result section was focused on the description of multiple clinical trials and their association with PD-L1. No expression-based analysis of PD-L1 in any dataset was available and without the knowledge of the different expression levels or even gene capture capabilities of PD-L1 in different GC subtypes, it is almost impossible to explore how PD-L1 would serve as a reliable biomarker for immunotherapy. 

Author Response

Thank you for the comments. 

We would like to emphasize that this article aims to review current and/or future predictive biomarkers for personalized immunotherapy in gastric cancer. Since this is a review article and is not a retrospective study, we did not include or review previous studies using datasets (expression or mutational profiles) for immunotherapy in gastric cancer. 

We instead reviewed clinical trials such as Checkmate 649 study which demonstrated that PD-L1 expression level can be used as predictive biomarkers for the treatment of GC patients. 

If the reviewer would prefer to change the title of our Review Article paper, I agree with that.

Reviewer 2 Report

The manuscript “Predictive biomarkers for personalized immunotherapy in gastric cancer” by Kim et al., is a review article, proposed to describe predictive biomarkers for immunotherapy with immune checkpoint inhibitors and help select the appropriate therapeutic approaches for patients with gastric cancer.

The authors proposed a list of predictive biomarkers and focused mainly on PD-L1 expression.

This article is well written and the different approaches are well organized.

I have just an idea in order to improve the presentation of the manuscript and to facilitate the reading. I recommend to the authors to add :

1) a table with the list of clinical assays related to the topic, with the characteristics of the assays

2) a table with the list of predictive biomarkers, the context and the applications.

Author Response

Thank you for the comments. 

According to your comments, we added figure 1, Table 1, and Table 2.